# Association of Dietary Inflammatory Index with Serum IL-6, IL-10, and CRP Concentration during Pregnancy

**DOI:** 10.3390/nu12092789

**Published:** 2020-09-11

**Authors:** Joanna Pieczyńska, Sylwia Płaczkowska, Lilla Pawlik-Sobecka, Izabela Kokot, Rafał Sozański, Halina Grajeta

**Affiliations:** 1Department of Food Science and Dietetics, Wroclaw Medical University, Borowska 211, 50-556 Wrocław, Poland; halina.grajeta@umed.wroc.pl; 2Department of Laboratory Diagnostics, Wroclaw Medical University, Borowska 211a, 50-556 Wrocław, Poland; sylwia.placzkowska@umed.wroc.pl (S.P.); izabela.kokot@umed.wroc.pl (I.K.); 3Department of Clinical Nursing, Wroclaw Medical University, K. Bartla 5, 51-618 Wrocław, Poland; lilla.pawlik-sobecka@umed.wroc.pl; 41st Department and Clinic of Gynaecology and Obstetrics, Wroclaw Medical University, T. Chałubińskiego 3, 50-368 Wrocław, Poland; rafal.sozanski@umed.wroc.pl

**Keywords:** dietary inflammatory index, maternal diet, interleukin, CRP

## Abstract

**Background**: The mother’s diet has a direct impact on fetal development and pregnancy, and can also be important in the course of the body’s inflammatory response. An anti-inflammatory diet can be a promising way to counter an excessive inflammatory response in pregnancy. **Objective**: The aim of the study was to examine the association between the dietary inflammatory index (DII) and the pregnant women’s serum interleukin 6 (IL-6) and 10 (IL-10) and C-reactive protein (CRP) concentration in the course of normal and complicated pregnancy. **Research Methods and Procedures**: The study included 45 Polish pregnant women recruited to the study. The DII, a literature-based dietary index to assess the inflammatory properties of diet, was estimated based on a seven-day 24-h recall and an food frequency questionnaire (FFQ) in each trimester of pregnancy. At the same time as the nutritional interviews, blood samples were collected for the determination of IL-6, IL-10, and CRP concentrations. The studied group was divided into subgroups with normal and complicated pregnancy and depending on the DII median. **Results**: With the development of pregnancy, the DII score slightly decreased in subsequent trimesters: −1.78 in the first trimester, −2.43 in the second trimester, and −2.71 in the third trimester (*p* = 0.092). Independent of the trimester of pregnancy and the occurrence of pregnancy complications, the DII score did not affect the differences in the serum concentrations of IL-6, IL-10, and CRP, with the exception of CRP level in the second trimester in women with complicated pregnancy (subgroup with DII < median had a lower CRP level than subgroup with DII > median). In the first and third trimesters, there was a weak but significant positive correlation between the DII score and CRP concentration. During the second trimester, in the group with normal pregnancy and DII below the median, a significant negative correlation between the DII score and the serum IL-6 and IL-10 concentration was noted as well as in the third trimester for IL-6. **Conclusion**: The anti-inflammatory potential of a pregnant woman’s diet increases slightly with pregnancy development; however, its value has no permanent significant association with the level of CRP, IL-6, and IL-10.

## 1. Introduction

Pregnancy is a physiological process that occurs in a woman’s body and consists of a series of immunobiological reactions. The woman’s immune system plays a particularly important role in pregnancy, protecting her body against pathogens and at the same time, allowing the acceptance of antigenically foreign sperm without starting an immune response. Placental tissue produces cytokines and hormones that are essential to the regulation of the feto-maternal unit. Recent studies suggested that in pregnant women, cytokines produced by Type 2 T helper (Th2) cells predominate over those produced by Type 1 T helper (Th1) cells, resulting in the maintenance of pregnancy. Th2 cells secrete interleukins 4, 5, 6, 10, and 13 and are associated with help for humoral immunity [1,2]. Immunological processes in pregnant women increase the concentration of cytokines in comparison to non-pregnant women, and for this reason, pregnancy is recognized as a “naturally inflammatory state” [3,4,5]. During pregnancy, three stages of inflammatory response can be distinguished, and during the second stage of pregnancy, there is a developing tolerance between the mother and the fetus. This fact is associated with changes in cytokine concentrations in the subsequent stages of pregnancy. The first stage of pregnancy is associated with inflammation, which is required for blastocyst implantation, then pro-inflammatory cytokine level increase. The second stage is characterized by an elevated anti-inflammatory cytokine level that is necessary for fetal growth. In the third stage, there is a change to an inflammatory immune state with a high pro-inflammatory cytokine level, which is necessary for labor and delivery [6,7,8,9].

Besides research on endocrinologic and infectious etiologies, a great deal of attention has currently been focused on the possible immunological causes of pregnancy complications. Excessive increase in inflammatory response in a pregnant woman’s organism is not beneficial for her and may result in complications of her health. In some cases, higher concentrations of some inflammatory markers are associated with pregnancy complications such as: premature fetal membrane rupture, premature labor, hypertension, and preeclampsia, considered to be one of the most common causes of prematurity [10,11,12,13].

The mother’s diet has a direct impact on fetal development and pregnancy and can also be important in the course of the body’s inflammatory response. The “Western-type” diet, rich in red meat, high-fat dairy products, and simple carbohydrates, may contribute to increasing the concentration of inflammatory mediators [14]. Studies show that in adults whose diets consist of large amounts of fruits, vegetables, filter plants, fish, poultry, and whole grain bread, the concentration of C-reactive protein (CRP) is lower than in people using other diets [15,16]. One study with pregnant women found that a diet rich in protein of animal origin has pro-inflammatory effects [17].

DII (Dietary Inflammatory Index) is an index that allows the assessment of a diet in terms of its pro- and anti-inflammatory effects on the body. Studies show correlations between pro-inflammatory DII values and increased CRP and IL-6 in non-pregnant adults [18,19]. The literature suggests that the knowledge of pregnant women regarding proper nutrition and the impact of nutrition on the course of pregnancy and the child’s health is inadequate [20,21]. The study by Sen et al., involving pregnant women and their children, shows that a maternal pro-inflammatory diet (DII above 0) can affect reduced birth weight of the child and problems with breastfeeding [22]. An anti-inflammatory diet can be a promising way to counter an excessive inflammatory response in pregnancy, especially in obese pregnant women. The increase in cytokine concentration is influenced not only by pregnancy but also by the amount of fat tissue in the pregnant woman; both of these factors individually contribute to a state of chronic inflammation. Maternal obesity in pregnancy is associated with an increase in cytokines, protein hormones, and the concentration of acute phase proteins [23,24]. Research indicates that women using a pro-inflammatory diet have an increased risk of premature delivery of offspring. Similarly, there was an association between this type of diet and a higher risk of caesarean section in obese women [25].

The aim of this study was to examine the association between the DII and pregnant women’s serum interleukin 6 (IL-6) and 10 (IL-10) and CRP concentration in the course of normal and complicated pregnancy.

## 2. Material and Methods

### 2.1. Participants

Forty-five pregnant women at different stages of pregnancy were qualified for the study. Subjects were recruited from several private obstetric clinics in the Lower Silesia region of Poland. The following exclusion criteria were applied: a multiple pregnancy, preexisting hypertension, diabetes mellitus, pregestational hypothyroidism, autoimmune diseases, recurrent cystitis, or present or previous cardiovascular disease. The Wroclaw Medical University Bioethics Committee KB-884/2012 approved the study. The occurrence of pregnancy complications such as gestational diabetes, gestational hypertension, gestational hypothyroidism, urinary tract infections, and anemia was based on medical records. The different number of subjects in the subsequent stages of pregnancy results from the fact that only 18 women participated in the study from the first to the third trimester; the remaining participants either joined later in the study or resigned from participation in the study before the third trimester. All participants of the study used mineral and vitamin supplements, with a diverse composition dedicated to pregnant women from the first trimester throughout the whole period of pregnancy. The pregnant ones in our study were informed on each visit, after a nutritional interview, about the food errors that they made and obtained nutritional advice appropriate to the pregnancy stage. The studied group characteristics are shown in Table 1.

### 2.2. Blood Samples

Blood samples from study participants were collected at three different timepoints across gestation, i.e., 8–14 weeks of gestation (T I), 18–24 weeks of gestation (T II), and 31–36 weeks of gestation (T III). Venous blood was drawn in the morning after an overnight fast. From blood samples, serum was obtained by centrifugation (2300× *g*, 15 min at 4 °C) within a maximum of 2 h after extraction. Thereafter, it was immediately aliquoted and stored at −80 °C for use in IL-6, IL-10, and hs-CRP measurements.

### 2.3. Biochemical Measurements

#### 2.3.1. Determination of IL-6 and IL-10 Concentrations

IL-6 and IL-10 concentrations were measured using an ELISA kit (DIACLONE Research, Besançon, France) with a minimum detectable dose for IL-6 −2 pg/mL and IL-10 −4.9 pg/mL. The assay employs the quantitative sandwich immunoassay technique, with a monoclonal antibody specific for IL-6 and IL-10 precoated onto a microplate. Standard curves were plotted for each of the interleukins using reference recombinant cytokines supplied by the manufacturer and the results read from the curves. Each serum sample was tested in duplicate in a blinded fashion.

#### 2.3.2. Determination of CRP Concentration

Serum CRP (hs-CRP) concentration was determined using the ultrasensitive immunoturbidimetric assay DiaSys (Holzheim, Germany) and analyzer Konelab 20i (Thermo Scientific, Waltham, MA, USA)

### 2.4. Dietary Inflammatory Index (DII)

The DII was developed by researchers at the University of South Carolina. Development and validation of the DII has been published previously [20,26]. In each trimester of pregnancy, one week before blood sample collection, dietary data were collected by trained dieticians individually for each study participant using the 24 h recall method from 7 consecutive days and a food frequency questionnaire (FFQ) to estimate their usual diet in particular trimesters, including food and supplements. In the 24 h recall, participants recorded both the consumed food and the amount of specific vitamin and mineral supplements, together with the name of the preparation. This allowed for an estimate of the individual nutrient intake from both food and supplements. The mean daily intake of energy and nutrients was computed from 24 h recalls from 7 consecutive days using a tailored computer program Dieta 5.0 and data provided by the Institute of Food and Nutrition.

Based on data obtained from 24 h recalls and the FFQ questionnaire, 37 food parameters (energy, carbohydrate, protein, fat, alcohol, fiber, cholesterol, saturated fat, mono-unsaturated fat, poly-unsaturated fat, omega-3 fatty acids, omega-6 fatty acids, niacin, thiamin, riboflavin, vitamin-B12, vitamin-B6, vitamin A, β-carotene, vitamin C, vitamin D, vitamin E, folic acid, iron, magnesium, zinc, selenium, flavan-3-ols, flavones, flavonols, flavonoids, anthocyanins, isoflavones, caffeine, onion, garlic, and tea) were selected from the 45-element DII database, which were then used for calculations. The individual dietary data of participants were transformed into a z-score by using the global mean and standard deviation of each food parameter provided by the DII database [26]. To minimize the effect of ’right skewing’ (a common occurrence with dietary data), this value was then converted to a centered percentile score, which was then multiplied by the respective inflammatory effect score of the food parameters (derived from a literature review and scoring of 1943 ’qualified’ articles) to obtain the subject’s food parameter-specific DII score. All of these food parameter-specific DII scores were then summed to create the overall DII score for every pregnant woman in the study in particular trimesters of pregnancy. The more positive scores are more pro-inflammatory, and the negative scores are more anti-inflammatory.

### 2.5. Statistical Analysis

Due to the small number of participants, in groups for individual maternal complications in the subsequent trimesters of pregnancy (Table 1), it was not possible to perform reliable statistical analyses; therefore, all complications were included as one group. To compare the IL-6, IL-10, and CRP concentrations and DII value between three trimesters, we used nonparametric analysis of variance with the Kruskal–Wallis test. Mann–Whitney U tests were used to compare the IL-6, IL-10, and CRP concentrations and DII value between < DII median value and > DII median value subgroups and normal and complicated pregnancy groups. Correlations were analyzed by Spearman’s rank correlation test. All statistical analyses were performed using Statistica for Windows, version 12.0 PL (StatSoft Inc., Tulsa, OK, USA). Two-tailed tests and a 5% level of significance were applied when applicable.

## 3. Results

### 3.1. Participant Baseline Characteristics 

In the case of more than half of the participants (50.7%), maternal age was in the range of 25–30 years, the mean age being 30.6 ± 5.4 years. As pregnancy progressed, the mean body weight in the following trimesters increased from 63.12 kg in T I to 69.32 kg in T II and 75.28 kg in T III. In addition, with regard to pregnancy complications, their percentage increased from 13.6% in the first trimester to 58.0% in the third trimester. The majority of the investigated participants were people with higher education, living in urban areas, and who had never smoked (Table 1).

### 3.2. DII Score and Serum CRP, IL-6, and IL-10 Levels during Pregnancy

With the development of pregnancy, the DII score decreased in subsequent trimesters. With the exception of the first trimester, lower DII values were observed in the groups with uncomplicated pregnancy as compared to those with complicated pregnancy. However, in both cases, these differences were not statistically significant. As regards the level of CRP, the highest concentration was observed in the second trimester, and the occurrence of pregnancy complications did not significantly affect its level; however, statistically significant differences were found between the levels in the first and second trimester. The concentration of IL-6 gradually increased with the development of pregnancy, and the difference between the levels in the first and third trimester was statistically significant. The lowest concentration of serum IL-10 was found in the first trimester, while the highest was in the third trimester. However, regardless of the trimester and the occurrence of pregnancy complications, the differences were not statistically significant (Table 2).

### 3.3. Comparison of Serum Concentrations of IL-6, IL-10, and CRP Depending on the Median Score of the DII during Pregnancy 

Irrespective of the trimester of pregnancy and the occurrence of pregnancy complications, the division of pregnant women into groups with DII score above and below the median did not influence the occurrence of statistically significant differences in the serum concentrations of IL-6, IL-10, and CRP, with the exception of CRP level in the second trimester in women with complicated pregnancy (subgroup with DII < median had a lower CRP level than subgroup with DII > median) (Table 3).

### 3.4. Correlation Analysis of DII Score with IL-6, IL-10, and CRP Levels in the Particular Trimesters 

The correlation analysis between the DII score and the levels of the studied interleukins and CRP was performed in each trimester of pregnancy in the subgroups: below and above median DII score. This analysis did not show an association between the studied factors. The exception was the first trimester, where in the subgroup with DII below the median, there was a statistically significant weak positive correlation between the DII score and the CRP concentration as well as in the third trimester (Table 4).

### 3.5. Correlation Analysis of DII Score with IL-6, IL-10, and CRP Levels in the Particular Trimesters among Pregnant Women with Normal Pregnancy

We investigated the correlation between maternal serum IL-6, IL-10, and CRP concentration and the DII score (in subgroups: above and below the median DII score) in particular trimesters among women with normal pregnancy. We found evidence of a strong negative correlation between DII score below median and IL-6 and IL-10 concentration in the second trimester as well as between DII score below median and IL-6 concentration in the third trimester (Table 5).

### 3.6. Correlation Analysis of DII Score with IL-6, IL-10, and CRP Levels in the Particular Trimesters among Pregnant Women with Complicated Pregnancy 

We estimated also the correlation between IL-6, IL-10, and CRP levels and the DII score (in subgroups: above and below the median DII score) in women with complicated pregnancy. Analysis of the correlation shows only one weak positive correlation between CRP level and DII score in the second trimester. For other analyses, maternal DII score did not correlate with cytokine and CRP maternal serum levels (Table 6).

## 4. Discussion

Proper nutrition of a pregnant woman is a key element for optimal fetal development. With the advancement of pregnancy, the pregnant woman’s requirement for nutrients, vitamins, minerals, and energy increases. Abnormalities in the diet can be associated with the occurrence of pregnancy complications and/or can disrupt the immune system.

In the presented study, the data analysis showed that with the development of pregnancy, the DII score slightly decreased in subsequent trimesters: −1.78 in the first trimester, −2.43 in the second trimester, and −2.71 in the third trimester. Independent of the trimester of pregnancy and the occurrence of pregnancy complications, the DII score did not affect the differences in the serum concentrations of IL-6, IL-10, and CRP, with the exception of CRP level in the second trimester in women with complicated pregnancy (subgroup with DII < median had a lower CRP level than subgroup with DII > median). In the first and third trimesters, there was a weak but significant positive correlation between the DII score and CRP concentration. During the second trimester, in the group with normal pregnancy and DII below the median, a significant negative correlation between the DII score and the serum IL-6 and IL-10 concentration was noted as well as in the third trimester for IL-6.

The study group consisted of 45 women in different stages of pregnancy. The vast majority of the study participants had a higher education (Table 1). The average pre-pregnancy BMI for subjects was 20.11 kg/m^2^, and this result reflects normal average body weight. Over 93% of the studied group were non-smokers (Table 1). Sen and colleagues [22] showed a positive relationship between pre-pregnancy BMI, smoking tobacco, and the DII level. In turn, having a higher education correlated with a lower DII score [22]. The characteristics of the studied group can explain why a very high percentage among the analyzed daily food rations had a DII score below 0. Only 13.7% of the analyzed daily food rations had a pro-inflammatory DII score. The obtained results ranged from −6.12 to 2.59 (Table 2) and the mean DII score in this study for all pregnant women was −1.90. Interestingly, the DII score obtained for the first trimester (−1.51) is very close to the median DII score calculated for ten European populations (−1.53) [27]. Sen and colleagues [22] analyzed nutritional data of over 1800 pregnant women, which allowed them to determine a DII score for this group. The average DII score obtained by them amounted to −2.56. In other studies, a very similar mean score was obtained for the pregnant women’s DII daily food rations (−2.6) [28]. The negative value of DII score in our study and the cited studies shows the anti-inflammatory potential of the diet of women expecting a child, and slight differences between the DII score in our study and other studies may result from the size of the group, differences in nutrition traditions, and the number of elements taken to calculate the DII. Comparing the DII score at different stages of pregnancy, an increase in the anti-inflammatory potential of the diet was observed along with the subsequent trimesters, although the differences were not statistically significant (Table 2). The pregnant women in our study were informed on each visit about the food errors that they had made and obtained nutritional advice appropriate to the pregnancy stage. Irregular meals were observed among the most common food errors in the studied pregnant women as well as too low consumption of vegetables and fruit, fish, and milk and too high consumption of highly processed products (data not shown). A lower DII score corresponding to gestational age may be the result of the increased awareness of pregnant women regarding proper nutrition and the use of vitamin/mineral supplements as the pregnancy progresses.

Although the median DII scores in the second and third trimester in the group with gestational complications were higher compared to the group with normal pregnancy, results analysis did not show any statistical differences between them (Table 2). High group homogeneity in terms of education, age, and the group’s residence could contribute to such results. Research suggest that older age, higher education level, and socioeconomic status are associated with healthier dietary patterns among pregnant women [29,30,31]. This information indicates that within a homogeneous group, the food choices will be very similar, which may be reflected in the DII score.

Few studies assessed the effect of the inflammatory nature of the diet, measured using DII, on the child’s development and the occurrence of pregnancy complications. In a study conducted by McCullough [25], pregnant women with a pro-inflammatory diet had a higher rate of preterm delivery among female newborns. In addition, a higher DII score was associated with a higher risk of caesarean section and a lower birth weight of the child among obese women [22,25]. There are also reports that suggest an adverse effect of a diet with pro-inflammatory potential in pregnancy on the occurrence of obesity in offspring in early childhood [28]. On the other hand, there was no correlation between DII values and the occurrence of such diseases as preeclampsia or impaired glucose tolerance during pregnancy [22].

We concluded that a DII score below or above median does not affect the differences in IL-6, IL-10, and CRP levels, regardless of the course of pregnancy, among the studied pregnant women, with the exception of CRP level in the second trimester in women with complicated pregnancy (Table 3). However, we obtained a significant positive correlation between the DII score and the CRP level in the third trimester in all studied subjects and in the second trimester in the group with complicated pregnancy (Table 4 and Table 6). The CRP protein is produced in the liver but can also be synthesized by the placenta and released into the maternal circulation. Perhaps this is one of the mechanisms underlying the increased CRP concentration in pregnant women [32]. Shivappa and colleagues’ study [33,34] has shown that a pro-inflammatory diet is associated with increases in CRP concentrations, possibly due to an elevated level of oxygen stress. This observation was confirmed in other studies involving healthy adults, although none of these studies were with pregnant women [15,35,36]. In a study conducted by Tabung and colleagues [36] with postmenopausal women, higher DII scores significantly predicted higher plasma concentrations of IL-6, tumor necrosis factor (TNF)-alpha, and CRP. Similarly, in two other studies [16,37], the authors observed a positive association between increasing DII score and IL-6 level. However, McCullough and colleagues [25] did not show the link between maternal E-DII (energy-adjusted dietary inflammatory index) score and circulating cytokines. We found a negative association between adherence to the pro-inflammatory diet (increasing DII score) and the IL-6 and IL-10 concentration among pregnant women with normal pregnancy, with a DII score below median in the second and third trimesters (Table 5). The demonstrated correlation for IL-6 is surprising and different from the reports of other authors. IL-6 concentration increases with the duration of pregnancy, with a significantly increased level observed during delivery [7]. Similarly, the expression of IL-10 during pregnancy is variable and depends on the duration of pregnancy. In addition, reduced expression of IL-10 may be an important mechanism associated with the preparation of the body for the onset of labor [6]. Perhaps the pregnant woman’s body strives to maintain homeostasis in the circulation of the studied cytokines and excessive stimulation of the inflammatory response through the diet disturbs this homeostasis. Different results may be due to different study target populations, such as pregnant women, compared to general adults. A lack of clear correlations in CRP, IL-6, and IL-10 levels in association to the DII score among pregnant women may be partially due to elevated inflammatory responses triggered by the progression of pregnancy or have been overcome by pharmacotherapy in cases of complicated pregnancies.

The study of the association of the inflammatory potential of diet using the DII with the concentration of inflammatory mediators as well as on the course of pregnancy among women expecting a baby is an issue that the scientific community has only been dealing with for the last several years. Therefore, the data on the discussed issues are poor in the world’s scientific literature. The results of the studies conducted so far are not unequivocal. On the one hand, modulating the inflammatory process through a properly balanced diet seems to be a promising tool in the fight against excessive inflammatory response during pregnancy; on the other hand, some studies have shown no association between a pregnant woman’s diet and her body’s immune response. It should also be remembered that the DII has been validated on the non-pregnant adult population and may not apply to pregnant women. Therefore, it is important to further explore the topic by conducting extensive research involving pregnant women.

### Limitations of the Study

The present study has several limitations. The markers of inflammation analyzed in our study are highly dependent on the treatment provided in the event of pregnancy complications. Pharmacotherapy was applied to all pregnant women with complicated pregnancy, which could have influenced the obtained results. Another study limitation was the inability to conduct a prospective analysis enabling the comparison of individual participant results across trimesters. Furthermore, the dietary advice given to study participants may have impacted on the obtained results. The DII is a literature-based assessment of dietary inflammatory potential. Conclusions are limited by the small amount of studies examining nutritional influences during pregnancy, with much of the background literature extrapolated from studies of non-pregnant adults. In addition, our cohort was relatively small and homogeneous in terms of education, place of residence, the use of supplements, and smoking.

## 5. Conclusions

This study provides an assessment of dietary inflammation potential in particular trimesters and its association with inflammatory markers during pregnancy. The anti-inflammatory potential of the pregnant woman’s diet increases slightly with pregnancy development; however, its value has no permanent significant relation to the level of CRP, IL-6, and IL-10. Future research is warranted to explore whether diet is a significant contributor to overall maternal inflammation and to demonstrate the potential impact of dietary interventions during pregnancy on maternal and offspring health.

## Figures and Tables

**Table 1 nutrients-12-02789-t001:** Baseline characteristics of pregnant women.

	T I *n* = 36	T II *n* = 41	T III *n* = 34
Pregnancy complications *n* (% of group)			
No	31 (86.1%)	18 (43.9%)	12 (35.3%)
Yes	5 (13.9%)	23 (56.1%)	22 (64.7%)
Hypothyroidism	2 (5.5%)	8 (20.4%)	7 (20.6%)
Gestational diabetes mellitus	2 (5.5%)	2 (4.9%)	5 (14.7%)
Urinary tract infections	1 (2.9%)	11 (26.8%)	6 (17.6%)
Hypertension	-	1(2.4%)	-
Anemia	-	1 (2.4%)	4 (11.8%)
Weight (kg)	63.12 ± 11.04	69.32 ± 12.06	75.28 ± 12.09
Pregnancy (weeks)	10.98 ± 2.97	22.24 ± 3.45	33.97 ± 2.46
Mean age (year) 30.6 ± 5.4			
Education *n* (% of group)			
Elementary school	1 (2.2%)		
High school	8 (17.7%)		
Higher education	36 (80.1%)		
Place of residence *n* (% of group)			
Urban	39 (86.6%)		
Rural	6 (13.4%)		
Smoking cigarettes *n* (% of group)			
Current smoker	2 (4.4%)		
Quit smoking	16 (35.5%)		
Never smoked	27 (60.1%)		

**Table 2 nutrients-12-02789-t002:** Median of DII score and serum CRP, IL-6, and IL-10 levels during pregnancy, median (min:max).

	T I				T II				T III				
	Total *n* = 36	Normal Pregnancy *n* = 31	Complicated Pregnancy *n* = 5	*p*	Total *n* = 41	Normal Pregnancy *n* = 18	Complicated Pregnancy *n* = 23	*p*	Total *n* = 34	Normal Pregnancy *n* = 12	Complicated Pregnancy *n* = 22	*p*	*p* for Trimesters
DII	−1.78 (−2.65:−0.40)	−1.58 (−2.64:−0.26)	−2.32 (−2.96:−2.04)	0.397	−2.43 (−3.40:−0.91)	−2.48 (−3.71:−1.55)	−1.94 (−3.23:−0.78)	0.563	−2.71 (−3.56:−1.27)	−2.90 (−3.81:−1.31)	−2.61 (−3.32:−1.27)	0.801	0.092
CRP (mg/L)	1.26 (0.47:2.61)	1.31 (0.44:2.55)	1.05 (0.85:4.30)	0.348	2.93 (1.74:5.72)	3.19 (2.31:5.72)	2.79 (1.56:6.01)	0.634	2.51 (1.27:3.66)	2.45 (0.84:2.90)	2.95 (1.40:5.40)	0.194	0.002
IL-6 (pg/mL)	6.10 (2.45:6.80)	6.00 (2.50:6.70)	6.50 (2.40:7.10)	0.714	6.60 (5.70:7.00)	6.60 (5.50:7.00)	6.20 (5.80:7.20)	0.875	6.85 (6.20:7.40)	6.85 (6.50:7.45)	6.75 (6.00:7.30)	0.665	0.021
IL-10 (pg/mL)	13.00 (8.40:14.35)	13.00 (8.60:14.40)	12.5 (6,.0:13.30)	0.436	13.60 (12.00:14.60)	13.7 (13.20:14.60)	13.50 (12.00:15.00)	0.927	13.70 (12.90:14,60)	14.00 (13.15:14.70)	13.60 (12.90:4.40)	0.745	0.240

DII: dietary inflammatory index; CRP: C-reactive protein.

**Table 3 nutrients-12-02789-t003:** Comparison of serum concentrations of IL-6, IL-10, and CRP depending on the median value of the DII score during pregnancy, median (min:max).

	T I			T II			T III		
	DII < Median	DII > Median	*p*	DII < Median	DII > Median	*p*	DII < Median	DII > Median	*p*
Total									
	Median= −1.78			Median = −2.48			Median = −2.71		
	*n* = 18	*n* = 18		*n* = 21	*n* = 20		*n* = 17	*n* = 17	
IL-6 (pg/mL)	6.25 (1.80:7.10)	5.55 (1.80:8.10)	0.635	6.45 (1.90:9.50)	6.60 (2.70:9.50)	0.705	6.60 (2.40:7.60)	6.90 (1.80:9.90)	0.309
IL-10 (pg/mL)	13.25 (5.00:16.60)	12.25 (5.00:22.05)	0.235	13.50 (5.40:60.00)	13.70 (5.20:16.20)	0.368	13.50 (5.50:16.90)	14.00 (4.30:16.40)	0.370
CRP (mg/L)	1.26 (0.40:12.37)	1.20 (0.14:8.55)	0.752	3.71 (0.27:17.49)	2.86 (0.21:9.58)	0.322	2.45 (0.35:5.58)	2.57 (0.20:15.96)	0.249
Normal pregnancy									
	Median = −1.58			Median = −2.53			Median = −2.80		
	*n* = 15	*n* = 16		*n* = 9	*n* = 9		*n* = 6	*n* = 6	
IL-6 (pg/mL)	6.30 (1.80:7.10)	4.30 (1.80:8.10)	0.243	6.70 (2.20:7.70)	6.60 (1.90:7.20)	0.596	6.55 (2.40:7.40)	7.05 (6.50:8.40)	0.262
IL-10 (pg/mL)	14.00 (5.00:16.60)	11.15 (5.60:22.05)	0.069	13.6 (5.60:15.10)	13.7 (5.20:15.40)	0.459	13.10 (5.50:15.40)	14.00 (13.20:14.80)	0.810
CRP (mg/L)	1.31 (0.25:12.37)	1.03 (0.24:5.96)	0.722	4.01 (2.33:13.89)	2.31 (0.21:7.72)	0.251	2.45 (0.35:3.07)	2.51 (0.67:5.68)	0.522
Complicated pregnancy									
	Median = −2.31			Median = −1.94			Median = −2.60		
	*n* = 3	*n* = 2		*n* = 12	*n* = 11		*n* = 11	*n* = 11	
IL-6 (pg/mL)	6.50 (2.40:7.10)	4.85 (2.40:7.30)	1.000	6.20 (3.40:9.50)	6.50 (2.70:6.50)	0.711	6.90 (3.10:7.70)	6.50 (1.80:9.90)	0.870
IL-10 (pg/mL)	12.5 (6.50:13.30)	9.90 (5.40:14.40)	0.773	13.30 (5.40:60.00)	13.70 (9.00:16.20)	0.805	13.50 (5.96:16.90)	14.00 (4.30:16.40)	0.646
CRP (mg/L)	0.85 (0.84:4.30)	5.01 (1.05:8.97)	0.386	1.56 (0.27:17.49)	5.27 (1.74:9.58	0.034	2.98 (0.81:5.58)	2.02 (0.20:15.96)	0.511

DII: dietary inflammatory index; CRP: C-reactive protein.

**Table 4 nutrients-12-02789-t004:** Correlation analysis of DII score with IL-6, IL-10, and CRP levels in the particular trimesters.

	DII I *n* = 36		DII I < Me *n* = 18		DII I > Me *n* = 18		DII II *n* = 41		DII II < Me *n* = 21		DII II > Me *n* = 20		DII III *n* = 34		DII III < Me *n* = 17		DII III > Me *n* = 17	
	r	*p*	r	*p*	r	*p*	r	*p*	r	*p*	r	*p*	r	*p*	r	*p*	r	*p*
IL-6	−0.089	0.831	−0.283	0.254	0.032	0.899	−0.176	0.271	−0.282	0.216	−0.138	0.563	0.012	0.948	−0.352	0.166	−0.274	0.287
IL-10	−0.189	0.269	−0.398	0.102	0.345	0.161	0.123	0.442	−0.152	0.509	0.171	0.471	0.182	0.304	−0.018	0.944	0.054	0.837
CRP	0.037	0.831	0.496	0.036	−0.170	0.499	0.122	0.446	−0.074	0.749	0.030	0.900	0.136	0.044	−0.115	0.658	0.000	1.000

DII: dietary inflammatory index; CRP: C-reactive protein.

**Table 5 nutrients-12-02789-t005:** Correlation analysis of DII score with IL-6, IL-10, and CRP levels in the particular trimesters among pregnant women with normal pregnancy.

	DII I *n* = 31		DII I < Me *n* = 15		DII I > Me *n* = 16		DII II *n* = 18		DII II < Me *n* = 9		DII II > Me *n* = 9		DII III *n* = 12		DII III < Me *n* = 6		DII III > Me *n* = 6	
	r	*p*	r	*p*	r	*p*	R	*p*	r	*p*	r	*p*	r	*p*	r	*p*	r	*p*
IL-6	−0.200	0.289	−0.408	0.131	0.040	0.884	−0.30	0.229	−0.869	0.023	0.365	0.333	0.014	0.965	−0.829	0.042	−0.543	0.266
IL-10	−0.308	0.092	−0.445	0.96	0.331	0.210	0.055	0.827	−0.733	0.024	0.133	0.732	−0.42	0.896	−0.429	0.396	−0.086	0.872
CRP	0.051	0.786	0.455	0.089	−0.012	0.965	−0.288	0.246	−0.194	0.617	0.133	0.732	−0.03	0.991	−0.657	0.156	−0.200	0.704

DII: dietary inflammatory index; CRP: C-reactive protein.

**Table 6 nutrients-12-02789-t006:** Correlation analysis of DII score with IL-6, IL-10, and CRP levels in the particular trimesters among pregnant women with complicated pregnancy.

	DII I *n* = 5		DII I < Me *n* = 3		DII I > Me *n* = 2		DII II *n* = 23		DII II < Me *n* = 12		DII II > Me *n* = 11		DII III *n* = 22		DII III < Me *n* = 11		DII III > Me *n* = 11	
	r	*p*	r	*p*	r	*p*	R	*p*	r	*p*	r	*p*	r	*p*	r	*p*	r	*p*
IL-6	0.564	0.322	1.000	-	-	-	−0.149	0.498	0.104	0.748	−0.458	0.157	0.020	0.930	0.018	0.958	−0.041	0.904
IL-10	0.300	0.624	0.500	0.667	-	-	0.200	0.361	0.302	0.340	0.338	0.309	0.314	0.154	0.334	0.316	0.354	0.285
CRP	0.600	0.284	0.500	0.667	-	-	0.415	0.049	0.116	0.720	−0.087	0.800	0.232	0.299	0.218	0.519	0.173	0.611

DII: dietary inflammatory index; CRP: C-reactive protein.

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
