# Peer review of "Association of Dietary Inflammatory Index with Serum IL-6, IL-10, and CRP Concentration during Pregnancy"

_nutrients, 2020, doi:10.3390/nu12092789_

Round 1

Reviewer 1 Report

I believe that the authors have done a good job in responding my comments and revising the manuscript.

Author Response

Reviewer 1

I believe that the authors have done a good job in responding my comments and revising the manuscript.

The authors are very grateful for the kind review. All grammatical and stylistic errors were corrected by an American English native speaker.

Reviewer 2 Report

The authors aimed to examine the associations between the dietary inflammatory index (DII) and the concentrations of IL-6, IL-10 and CRP in pregnant women of different pregnancy stages. The pregnant women were recruited in the first, second and third trimesters of pregnancy, therefore this is not a prospective but a cross-sectional design. The authors found that the DII scores were lowest (anti-inflammatory) in the third trimester compared to the first trimester. The DII was not consistently associated with inflammatory markers, but in the first trimester, among women with a DII score < median score, the DII was significantly correlated with CRP levels. CRP levels were also correlated with the DII scores in the third trimester. In women with pregnancy complications, the CRP levels were highest for women that had a DII score > median score.  Overall, this paper is interesting, however there is maybe too many results presented, and not all are relevant. The topic is novel and not many studies assessed the inflammatory potential of the diet during pregnancy. I would recommend an extensive grammar and syntax editing by a native English speaker. The manuscript is very difficult to read as is.

MAJOR COMMENTS

  1. In the introduction, it is important to add references at the end of sentences that mention previous literature or that simply state general affirmations (e.g. at the end of: « Besides research on endocrinologic and infectious etiologies, currently a great deal of attention has been focused on the possible immunological causes of pregnancy complications. ») There needs to be a reference to back-up the statements made by the authors.
  2. The introduction is rather long and not very fluid. I would suggest the authors try to clearly state what are the gaps in the literature and why their study is relevant/necessary.
  3. It is not clear if the authors used intakes from supplements for their calculation of the DII scores. I would suggest the authors clarify that point because it is not clear throughout the manuscript.
  4. In general, there are a lot of results that are not relevant and difficult to understand. I do not understand the necessity to divide women on the basis of whether they are below or over the median DII score, I believe it is just more confusing. I understand that maybe the authors wanted to ‘find’ and present results that were statistically significant, but I believe it just makes the manuscript more difficult to read. It is acceptable to mention that not many statistically significant correlations and differences were observed.

MINOR COMMENTS

  1. I would suggest using the term ‘normal’ instead of ‘physiological’ pregnancy and the word ‘association’ instead of ‘relation’ or ‘relationship’ throughout the manuscript.
  2. Abstract and introduction. I suggest using the term ‘counter an excessive inflammatory response’ instead of ‘fight an excessive […]’.
  3. Abstract and results. Please further develop on the following results: « DII score didn’t influence on differences in the serum concentrations of IL-6, IL-10 and CRP with the exception of CRP level in the second trimester in women with complicated pregnancy ». Does this mean that serum concentrations of CRP in the 2nd trimester differed significantly according to the DII score? If so, please specify or make it clearer in the abstract as well as in the Results section.
  4. Introduction. It would be relevant to mention in the introduction that pregnancy is recognize as a ‘naturally inflammatory state’, but that in some cases, higher concentrations of some inflammatory markers are associated with some pregnancy complications (preeclampsia, gestational diabetes, preterm labor, etc.).
  5. Introduction. Please rephrase the following sentence and provide a reference, as it is difficult to understand: « […] and at the same time allowing the adoption of antigenically foreign sperm without starting an immune response. » I am not sure what the authors mean by ‘adoption’?
  6. Introduction. Did the authors mean ‘Type 2 T helper (Th2) predominate over Type 1 T helper’?
  7. Introduction. « On the basis of studies of placenta samples taken from pregnant women during different periods of pregnancy [..] » were those studies that included healthy pregnant women or pregnant women with complications? Please specify.
  8. Introduction. What does ‘the correctness of its course’ means when talking about pregnancy? Please use a more appropriate term.
  9. Introduction. I would refrain from saying ‘Studies show that […]’ when only one study is referenced. Please provide more than one reference if the plural form is used or rephrase the sentence by saying ‘one study found that…’, instead of ‘studies confirmed’. This was a recurrent issue in the introduction.
  10. Introduction. Please specify whether the associations between the DII and IL-6/CRP concentrations were observed in pregnant or non-pregnant adults.
  11. Introduction. Please clarify what is meant by ‘rational nutrition’. I would suggest the authors use another term.
  12. Introduction. Why mention women with obesity at the end of the introduction when the weight status of pregnant women was not discussed at all in the introduction? Please remove that part of the sentence or include some background information on obese women in the introduction.
  13. Methods. Please specify in the first sentence of the Methods section when pregnant women were recruited (first, second or third trimester)?
  14. Methods. Please try to rephrase the section on the calculation of the DII, as it currently looks like plagiarism. It is a very complex dietary index, so perhaps try to explain what ‘linked to the world database’ means? Maybe say that the individual dietary data of participants were transform into a z-score by using the global mean and standard deviation of each food parameter, that were provided by the DII database.
  15. Results. Were the U Mann–Whitney tests used to assess differences in DII scores and inflammatory markers? If so, I believe the authors should use the means and not the medians to do that analysis. Please modify your analyses/results accordingly.
  16. Results. Please rephrase the first sentence of the subsection ‘Participant baseline characteristics’.
  17. Results. What do the authors mean by « […] however, statistically significant differences were found between the results in the first and second trimester. »? Does this refer to differences between the levels of CRP in the first vs. the second trimester? If so, use the term ‘levels’ instead of ‘results’. The same comment applies for the following sentence on IL-6.
  18. Results. Please clarify what analysis is this: « Analysis of the correlation in the pregnant women group characterized by the DII score above and below the median, regardless of the trimester of pregnancy, did not show statistically significant dependences. » Do the authors mean that, in each ‘DII group’, they analyzed the correlations between DII scores and inflammatory markers? If so, make it clearer than how it is written currently.
  19. Results. Please rephrase the whole 3.5 and 3.6 subsections, it is very difficult to understand as it is written right now. Also, it is not clear if the authors report a correlation analysis or a U Mann–Whitney test. Please use the correct term.
  20. Discussion. Please provide a summary of the most important results at the beginning of the discussion.
  21. Discussion. Please rephrase the following sentence: « Education most of them was on academic level. »
  22. Discussion. Please further explain the following: « The pregnant ones in our study were informed on the first visit about the food errors they were made and they were fed with nutritional education. » What does this mean? No nutritional education was mentioned in the Methods section. Do you mean that pregnant women were given nutritional advice in the first trimester, thus this could have influence their diet for the rest of the pregnancy? Please clarify.
  23. Discussion. I would suggest mentioning that the decrease in the DII scores was not statistically significant.
  24. Discussion. « Lower DII score corresponding to gestational age may be the result of increased awareness of pregnant women regarding proper nutrition and using vitamin-mineral supplements as the pregnancy progresses. »
  25. Discussion. « We concluded that the DII score does not affect the differences in IL-6, IL-10 and CRP levels regardless of the course of pregnancy […] » I am not sure if those results are in line with the analyses that were performed, therefore I don’t think it can be one of the conclusions of this article. I believe the authors assessed the differences in IL-6, IL-10 and CRP levels according to the median DII score. They did not analyze whether the increase in those markers was influenced by the DII score.
  26. Discussion. The limitations of the study should be included in the Discussion section. Also, another limitation of the study is that prospective analyses were not possible (e.g. to assess the change in DII scores throughout pregnancy) because the authors included women that were already in their second and third trimesters. Therefore, women were not compared to themselves across trimesters. This should be added as a limitation.

Author Response

Reviewer 2

The authors aimed to examine the associations between the dietary inflammatory index (DII) and the concentrations of IL-6, IL-10 and CRP in pregnant women of different pregnancy stages. The pregnant women were recruited in the first, second and third trimesters of pregnancy, therefore this is not a prospective but a cross-sectional design. The authors found that the DII scores were lowest (anti-inflammatory) in the third trimester compared to the first trimester. The DII was not consistently associated with inflammatory markers, but in the first trimester, among women with a DII score < median score, the DII was significantly correlated with CRP levels. CRP levels were also correlated with the DII scores in the third trimester. In women with pregnancy complications, the CRP levels were highest for women that had a DII score > median score.  Overall, this paper is interesting, however there is maybe too many results presented, and not all are relevant. The topic is novel and not many studies assessed the inflammatory potential of the diet during pregnancy. I would recommend an extensive grammar and syntax editing by a native English speaker. The manuscript is very difficult to read as is.

[Response]: We appreciate the reviewer’s comments. The authors revised the manuscript in accordance with all comments made by the reviewer. All corrections made in the manuscript text are marked in red. All grammatical and stylistic errors were corrected by an American English native speaker.

MAJOR COMMENTS

  1. In the introduction, it is important to add references at the end of sentences that mention previous literature or that simply state general affirmations (e.g. at the end of: « Besides research on endocrinologic and infectious etiologies, currently a great deal of attention has been focused on the possible immunological causes of pregnancy complications. ») There needs to be a reference to back-up the statements made by the authors.

[Response]: We appreciate the reviewer’s comments. As suggested by the reviewer, the Introduction section has been supplemented with adequate literature items

Mor, G., Aldo, P., & Alvero, A. B. (2017). The unique immunological and microbial aspects of pregnancy. Nature Reviews Immunology, 17(8), 469.

Eghbal‐Fard, S., Yousefi, M., Heydarlou, H., Ahmadi, M., Taghavi, S., Movasaghpour, A., ... & Rikhtegar, R. (2019). The imbalance of Th17/Treg axis involved in the pathogenesis of preeclampsia. Journal of Cellular Physiology, 234(4), 5106-5116.

Raghupathy, R., & Kalinka, J. (2008). Cytokine imbalance in pregnancy complications and its modulation. Front Biosci, 13(1), 985-94.

Ragsdale HB, Kuzawa CW,Borja JB, Avila JL, McDade TW. Regulation of inflammation during gestation and birth outcomes:Inflammatory cytokine balance predicts birth weight and length. Am J Hum Biol. 2019;31:e23245.https://doi.org/10.1002/ajhb.23245

Teng, Y. C., Lin, Q. D., Lin, J. H., Ding, C. W., & Zuo, Y. (2009). Coagulation and fibrinolysis related cytokine imbalance in preeclampsia: the role of placental trophoblasts. Journal of perinatal medicine, 37(4), 343-348.

Agudelo, O. M., Aristizabal, B. H., Yanow, S. K., Arango, E., Carmona-Fonseca, J., & Maestre, A. (2014). Submicroscopic infection of placenta by Plasmodium produces Th1/Th2 cytokine imbalance, inflammation and hypoxia in women from north-west Colombia. Malaria Journal, 13(1), 122.

  1. The introduction is rather long and not very fluid. I would suggest the authors try to clearly state what are the gaps in the literature and why their study is relevant/necessary.

[Response]: We appreciate the reviewer’s comments. As suggested by the reviewer, the Introduction section has been shortened in the section on cytokines (p.2 l. 47-56)

“Immunological processes in pregnant women increase the concentration of cytokines in comparison to non-pregnant women and for this reason pregnancy is recognize as a ”naturally inflammatory state” [3, 4, 5]. During pregnancy, three stages of inflammatory response can be distinguished, and during the second stage of pregnancy, there is a developing tolerance between the mother and the fetus. This fact is associated with changes in cytokine concentrations in the subsequent stages of pregnancy [6-9].

The increase in cytokine concentration is influenced not only by pregnancy but also by the amount of fat tissue in the pregnant woman, both of these factors individually contribute to a state of chronic inflammation. Maternal obesity in pregnancy is associated with an increase in cytokines, protein hormones and acute phase proteins concentration [10, 11].”

  1. It is not clear if the authors used intakes from supplements for their calculation of the DII scores. I would suggest the authors clarify that point because it is not clear throughout the manuscript.

[Response]: We appreciate the reviewer’s comments. As suggested by the reviewer, the authors completed the Dietary Inflammatory Index (DII) section with missing information (p.4 l.130-132)

„In the 24-h recall, participants recorded both the consumed food and the amount of specific vitamin and mineral supplements together with the name of the preparation. This allowed for an estimate of the individual nutrient intake from both food and supplements.”

  1. In general, there are a lot of results that are not relevant and difficult to understand. I do not understand the necessity to divide women on the basis of whether they are below or over the median DII score, I believe it is just more confusing. I understand that maybe the authors wanted to ‘find’ and present results that were statistically significant, but I believe it just makes the manuscript more difficult to read. It is acceptable to mention that not many statistically significant correlations and differences were observed.

 [Response]: We appreciate the reviewer’s comments.

The aim of the study was to examine the association between the DII score and the pregnant women serum IL-6 and IL-10 and CRP concentration in the pregnancy course. The authors wanted to present the DII score as the "main character" of the study. Initially, we assumed that in each trimester of pregnancy we would be able to distinguish two groups: with a DII score above 0 (pro-inflammatory potential) and with a DII score below 0 (anti-inflammatory potential). Unfortunately, of all the individual studied diets, only a few had pro-inflammatory potential. Due to the above, we decided to refer the obtained results to the median DII. In the previous version of the manuscript, in order to limit the number of presented results, we presented only statistically significant correlations. However, at the urgent request of one of the reviewers, we presented all the correlations obtained during the calculations. At this point we feel "torn" as to which of the reviewers should agree.

MINOR COMMENTS

  1. I would suggest using the term ‘normal’ instead of ‘physiological’ pregnancy and the word ‘association’ instead of ‘relation’ or ‘relationship’ throughout the manuscript.

 [Response]: We appreciate the reviewer’s comments.

The authors have replaced the indicated words throughout the text of the manuscript

  1. Abstract and introduction. I suggest using the term ‘counter an excessive inflammatory response’ instead of ‘fight an excessive […]’.

 [Response]: We appreciate the reviewer’s comments.

The authors have replaced the indicated words throughout the text of the manuscript

  1. Abstract and results. Please further develop on the following results: « DII score didn’t influence on differences in the serum concentrations of IL-6, IL-10 and CRP with the exception of CRP level in the second trimester in women with complicated pregnancy ». Does this mean that serum concentrations of CRP in the 2nd trimester differed significantly according to the DII score? If so, please specify or make it clearer in the abstract as well as in the Results section.

 [Response]: We appreciate the reviewer’s comments.

The authors have improved the description of the results. 

„Independent of the trimester of pregnancy and the occurrence of pregnancy complications, the DII score didn’t cause differences in the serum concentrations of IL-6, IL-10 and CRP with the exception of CRP level in the second trimester in women with complicated pregnancy (subgroup with DII < median vs subgroup with DII > median).”

  1. Introduction. It would be relevant to mention in the introduction that pregnancy is recognize as a ‘naturally inflammatory state’, but that in some cases, higher concentrations of some inflammatory markers are associated with some pregnancy complications (preeclampsia, gestational diabetes, preterm labor, etc.).

[Response]: We appreciate the reviewer’s comments. As suggested by the reviewer, the authors improved this part of the Introduction section as follows:

“ Immunological processes in pregnant women increase the concentration of cytokines in comparison to non-pregnant, women and for this reason, pregnancy is recognized as a ”naturally inflammatory state””

“In some cases, higher concentrations of some inflammatory markers are associated with pregnancy complications  such as: premature fetal membrane rapture, premature labor, hypertension and preeclampsia, considered to be one of the most common causes of prematurity [12-15].”

  1. Introduction. Please rephrase the following sentence and provide a reference, as it is difficult to understand: « […] and at the same time allowing the adoption of antigenically foreign sperm without starting an immune response. » I am not sure what the authors mean by ‘adoption’?

[Response]: We appreciate the reviewer’s comments. As suggested by the reviewer, the authors improved this word in the Introduction section as follows:

„The woman's immune system plays a particularly important role in pregnancy, protecting her body against pathogens and at the same time allowing the acceptance of antigenically foreign sperm without starting an immune response.”

  1. Introduction. Did the authors mean ‘Type 2 T helper (Th2) predominate over Type 1 T helper’?

[Response]: We appreciate the reviewer’s comments. The authors corrected this error.

  1. Introduction. « On the basis of studies of placenta samples taken from pregnant women during different periods of pregnancy [..] » were those studies that included healthy pregnant women or pregnant women with complications? Please specify.

[Response]: We appreciate the reviewer’s comments. This piece of text has been removed

  1. Introduction. What does ‘the correctness of its course’ means when talking about pregnancy? Please use a more appropriate term.

[Response]: We appreciate the reviewer’s comments. This piece of text has been removed

  1. Introduction. I would refrain from saying ‘Studies show that […]’ when only one study is referenced. Please provide more than one reference if the plural form is used or rephrase the sentence by saying ‘one study found that…’, instead of ‘studies confirmed’. This was a recurrent issue in the introduction.

[Response]: We appreciate the reviewer’s comments. The authors corrected these erros as follows:

One study with pregnant women found that a diet rich in protein of animal origin has pro-inflammatory effects [22].

“Sen et al. study, involving pregnant women and their children, show that the use of a pro-inflammatory diet during pregnancy (DII above 0) can cause reduced birth weight of the child and problems with breastfeeding [24].”

  1. Introduction. Please specify whether the associations between the DII and IL-6/CRP concentrations were observed in pregnant or non-pregnant adults.

[Response]: We appreciate the reviewer’s comments. The authors corrected this sentence as follows:

“Studies show correlations between proinflammatory DII values ​​and increased CRP and IL-6 in non-pregnant adults [24; 25].”

  1. Introduction. Please clarify what is meant by ‘rational nutrition’. I would suggest the authors use another term.

[Response]: We appreciate the reviewer’s comments. The authors corrected this sentence as follows:

„The literature suggests that the knowledge of pregnant women regarding proper nutrition and the impact of nutrition on the course of pregnancy and the child's health is inadequate”

  1. Introduction. Why mention women with obesity at the end of the introduction when the weight status of pregnant women was not discussed at all in the introduction? Please remove that part of the sentence or include some background information on obese women in the introduction.

[Response]: We appreciate the reviewer’s comments. As suggested by the reviewer, the authors completed the Introduction section on  obese pregnant women information as follows:

“The increase in cytokine concentration is influenced not only by pregnancy but also by the amount of fat tissue in the pregnant woman; both of these factors individually contribute to a state of chronic inflammation. Maternal obesity in pregnancy is associated with an increase in cytokines, protein hormones and the concentration of acute phase proteins [10, 11]”

  1. Methods. Please specify in the first sentence of the Methods section when pregnant women were recruited (first, second or third trimester)?

[Response]: We appreciate the reviewer’s comments. The authors corrected this sentence as follows:

„Fourty-five subjects involved in this study were recruited at  several obstetric private clinics in the Lower Silesia region of Poland at different pregnancy stages.”

  1. Methods. Please try to rephrase the section on the calculation of the DII, as it currently looks like plagiarism. It is a very complex dietary index, so perhaps try to explain what ‘linked to the world database’ means? Maybe say that the individual dietary data of participants were transform into a z-score by using the global mean and standard deviation of each food parameter, that were provided by the DII database.

[Response]: We appreciate the reviewer’s comments. The authors corrected this sentence as follows:

„The individual dietary data of participants were transformed into a z-score by using the global mean and standard deviation of each food parameter provided by the DII database [26].”

  1. Results. Were the U Mann–Whitney tests used to assess differences in DII scores and inflammatory markers? If so, I believe the authors should use the means and not the medians to do that analysis. Please modify your analyses/results accordingly.

 [Response]: We appreciate the reviewer’s comments.

As the Shapiro-Wilk test showed an abnormal distribution of results, the non-parametric Mann-Whitney U test was used for the comparative analysis. According to statistics textbooks, in this type of tests the median reflects the average value of the results better than the mean.

Nachar, N. (2008). The Mann-Whitney U: A test for assessing whether two independent samples come from the same distribution. Tutorials in quantitative Methods for Psychology, 4(1), 13-20.

MacFarland, T. W., & Yates, J. M. (2016). Mann–whitney u test. In Introduction to nonparametric statistics for the biological sciences using R (pp. 103-132). Springer, Cham.

  1. Results. Please rephrase the first sentence of the subsection ‘Participant baseline characteristics’.

[Response]: We appreciate the reviewer’s comments. The authors rephrase this sentence as follows:

„In the case of more than half of the participants (50.7%), the maternal age was in the range of 25-30 years, the mean age being 30.6±5.4 years.”

  1. Results. What do the authors mean by « […] however, statistically significant differences were found between the results in the first and second trimester. »? Does this refer to differences between the levels of CRP in the first vs. the second trimester? If so, use the term ‘levels’ instead of ‘results’. The same comment applies for the following sentence on IL-6.

 [Response]: We appreciate the reviewer’s comments.

The authors have replaced the indicated words as follows:

„As regards the level of CRP, the highest concentration was observed in the second trimester, and the occurrence of pregnancy complications did not significantly affect its level, however, statistically significant differences were found between the levels in the first and second trimester. The concentration of IL-6 gradually increased with the development of pregnancy, and the difference between the levels in the first and third trimester was statistically significant.”

  1. Results. Please clarify what analysis is this: « Analysis of the correlation in the pregnant women group characterized by the DII score above and below the median, regardless of the trimester of pregnancy, did not show statistically significant dependences. » Do the authors mean that, in each ‘DII group’, they analyzed the correlations between DII scores and inflammatory markers? If so, make it clearer than how it is written currently.

[Response]: We appreciate the reviewer’s comments.

The authors have improved the description of the results as follows:

„The correlation analysis between the DII score and the levels of the studied interleukins and CRP was performed in each trimester of pregnancy in the subgroups: below and above median DII score. This analysis did not show association between studied factors.”

  1. Results. Please rephrase the whole 3.5 and 3.6 subsections, it is very difficult to understand as it is written right now. Also, it is not clear if the authors report a correlation analysis or a U Mann–Whitney test. Please use the correct term.

[Response]: We appreciate the reviewer’s comments.

The authors have improved the description of the results as follows:

3.5

We investigated the correlation between maternal circulating  two cytokines and CRP and the DII score (in subgroups: above and below the median DII score) in particular trimesters among women with normal pregnancy. We found evidence of a strong negative correlation between DII score below median and IL-6 and IL-10 concentration in the second trimester as well as between DII score below median and IL-6 concentration in the third trimester.

3.6

We estimated also the correlation between IL-6, IL-10 and CRP levels and the DII score (in subgroups: above and below the median DII score) in women with complicated pregnancy. Analysis of the correlation shows only one weak positive correlation between CRP level and DII score in the second trimester. For other analyses, maternal DII score did not correlate with cytokine and CRP maternal serum levels.

  1. Discussion. Please provide a summary of the most important results at the beginning of the discussion.

[Response]: We appreciate the reviewer’s comments.

The authors have improved the Discussion section as follows:

In the presented study, the data analysis showed that with the development of pregnancy, the score of DII slightly decreased in subsequent trimesters: -1.78 in the first trimester, -2.43 in the second trimester and -2.71 in the third trimester. Independent of the trimester of pregnancy and the occurrence of pregnancy complications, the DII score didn’t cause differences in the serum concentrations of IL-6, IL-10 and CRP with the exception of CRP level in the second trimester in women with complicated pregnancy (subgroup with DII < median vs subgroup with DII > median). In the first and third trimester, there was waek, but significant, positive correlation between DII score and CRP concentration. During the second trimester in the group with normal pregnancy and DII below the median, a significant negative correlation between the DII score and the serum IL-6 and IL-10 concentration was noted as well as in third trimester for IL-6.

  1. Discussion. Please rephrase the following sentence: « Education most of them was on academic level. »

[Response]: We appreciate the reviewer’s comments.

The authors have rephrase the following sentence:

The vast majority of the study participants had a higher education.

  1. Discussion. Please further explain the following: « The pregnant ones in our study were informed on the first visit about the food errors they were made and they were fed with nutritional education. » What does this mean? No nutritional education was mentioned in the Methods section. Do you mean that pregnant women were given nutritional advice in the first trimester, thus this could have influence their diet for the rest of the pregnancy? Please clarify.

[Response]: We appreciate the reviewer’s comments.

The authors have improved the Discussion section as follows:

„The pregnant ones in our study were informed on each visit, after a nutritional interview, about the food errors that they made and obtained nutritional advice appropriate to the pregnancy stage.”

Discussion. I would suggest mentioning that the decrease in the DII scores was not statistically significant.

[Response]: We appreciate the reviewer’s comments.

The authors have improved the Discussion section as follows:

„Comparing the DII score at different stages of pregnancy, an increase in the anti-inflammatory potential of the diet was observed along with subsequent trimesters,  although the differences were not statistically significant (Table 2).”

  1. Discussion. « Lower DII score corresponding to gestational age may be the result of increased awareness of pregnant women regarding proper nutrition and using vitamin-mineral supplements as the pregnancy progresses. »

[Response]: We appreciate the reviewer’s comments.

The authors have improved the sentence as follows:

„Lower DII score corresponding to gestational age may be the result of increased awareness of pregnant women regarding proper nutrition and usign vitamin-mineral supplements as the pregnancy progresses.”

  1. Discussion. « We concluded that the DII score does not affect the differences in IL-6, IL-10 and CRP levels regardless of the course of pregnancy […] » I am not sure if those results are in line with the analyses that were performed, therefore I don’t think it can be one of the conclusions of this article. I believe the authors assessed the differences in IL-6, IL-10 and CRP levels according to the median DII score. They did not analyze whether the increase in those markers was influenced by the DII score.

[Response]: We appreciate the reviewer’s comments.

The authors have improved the Discussion section as follows:

„We concluded that the DII score below or above median does not affect the differences in IL-6, IL-10 and CRP levels regardless of the course of pregnancy among the studied pregnant womens with the exception of CRP level in the second trimester in women with complicated pregnancy (Tab. 3).”

  1. Discussion. The limitations of the study should be included in the Discussion section. Also, another limitation of the study is that prospective analyses were not possible (e.g. to assess the change in DII scores throughout pregnancy) because the authors included women that were already in their second and third trimesters. Therefore, women were not compared to themselves across trimesters. This should be added as a limitation.

[Response]: We appreciate the reviewer’s comments.

The authors have included the linitations in the Discussion section and improved as follows:

„Another study limitation was the inability to conduct a prospective analysis enabling the comparison of individual participant results across trimesters.”

Reviewer 3 Report

In referring to my comment related to calculating the DII with supplements vs. without, it is not correct that the DII was based on nutrients from all sources. In working with the developers very closely, early work separated out the DII with supplements and without. In fact, the early validation work was based on diet only. I think the authors are missing a key opportunity to examine the associations excluding the supplements. In doing so, you can start to understand the impact of supplements on the outcomes being studied depending on if you see the same or different associations when comparing results of the DII with and without nutrients.  

With the way the changes were noted in the response, it is hard to tell what changes were made for what statements. For example, I can't tell if changes were made for the following previous comments:

  1. Somewhere in the paper, can you provide the range of values of DII scores. I am seeing mean values over 1, but your median was -2 or so. This doesn’t make sense.
  2. I am not sure the figure helps for the correlations. You are only showed selected correlations. It would be better to show a table with all correlations you looked at. Again, some of the lack of significance may be due to a small sample size. High correlations are still high correlations even if they are not statistically significant. Readers need to be able to see that.
  3. Lastly, one reason you may be seeing a lack of significance with the DII is because you included supplements in the table. Supplements are loaded with many anti-inflammatory micronutrients and minerals. When supplements are included in the DII calculation, sometimes the values for those parameters get maxed out. In other words, there is a maximum value for the DII food parameters. If a large percentage of women are nearing those thresholds, that could cause a lack of significance. It also is important to note that absorption of micronutrients from diet and supplements are not the same. Again, I think it would be vital to run the same set of analyses with the DII from food intake only to see if your results differ.

I disagree with the authors statement about mentioning biological mechanisms for the findings is not appropriate for this work. One of Sir Austin Bradford Hill's causality criteria which is a staple in epidemiology is biological plausibility. Many DII papers provide plausible explanation for their findings. This helps to drive future research and to open the door for collaborations. You do not need to go into exhaustive detail, because as you say, you didn't investigate that. But, to mention a few mechanisms would be helpful. 

Lastly, the manuscript that was uploaded does not have any indications of where changes were made. The authors noted changes that were made, but unfortunately, it is not possible to easily find such changes. 

Author Response

Reviewer 3

Comments and Suggestions for Authors

In referring to my comment related to calculating the DII with supplements vs. without, it is not correct that the DII was based on nutrients from all sources. In working with the developers very closely, early work separated out the DII with supplements and without. In fact, the early validation work was based on diet only. I think the authors are missing a key opportunity to examine the associations excluding the supplements. In doing so, you can start to understand the impact of supplements on the outcomes being studied depending on if you see the same or different associations when comparing results of the DII with and without nutrients.  

[Response]: We appreciate the reviewer’s comments.

The authors do belive that the nutrients intake from supplements influences the inflammatory potential of the diet, therefore it should be taken into account in the calculation of DII, despite poor bioavailability. This belief is confirmed by the validation study quoted below, where the co-authors are the developers of the DII index (Shivappa, Hurley and Hebert).

„The FFQ included questions on nutritional suplement use for 15 nutrient components of the DII; namely, iron, magne-sium, niacin, riboflavin, selenium, thiamine,b-carotene, zinc, folicacid, and vitamins A, C, D, E, B6, and B12.” Tabung, F. K., Steck, S. E., Zhang, J., Ma, Y., Liese, A. D., Agalliu, I., ... & Martin, L. W. (2015). Construct validation of the dietary inflammatory index among postmenopausal women. Annals of epidemiology, 25(6), 398-405.

Nevertheless, the reviewer's suggestion inspired the authors to conduct a study in which the relationship between the DII score, calculated with regard to supplements and without it, and an immune response will be analyzed. We will conduct the study on 18 pregnant women who participated in the study from the first to the third trimester.

With the way the changes were noted in the response, it is hard to tell what changes were made for what statements. For example, I can't tell if changes were made for the following previous comments:

  1. Somewhere in the paper, can you provide the range of values of DII scores. I am seeing mean values over 1, but your median was -2 or so. This doesn’t make sense.

[Response]: We appreciate the reviewer’s comments. We have revised the Discussion  section:

- DII -1.90 is the overall mean for all three trimesters. The corrected sentence reads as follows " The obtained results ranged from -6.12 to 2.59 (Table 2) and the mean DII score in this study for all pregnant women was -1.90.” p. 11, l. 230-240

  1. I am not sure the figure helps for the correlations. You are only showed selected correlations. It would be better to show a table with all correlations you looked at. Again, some of the lack of significance may be due to a small sample size. High correlations are still high correlations even if they are not statistically significant. Readers need to be able to see that.

[Response]: We appreciate the reviewer’s comments.

- following the Reviewer's recommendation, all tables present p as a numerical value

- the authors presented all the correlation results as tables, not figures (tab. 4, 5, 6) – p. 10, 11, l. 212-216.

  1. Lastly, one reason you may be seeing a lack of significance with the DII is because you included supplements in the table. Supplements are loaded with many anti-inflammatory micronutrients and minerals. When supplements are included in the DII calculation, sometimes the values for those parameters get maxed out. In other words, there is a maximum value for the DII food parameters. If a large percentage of women are nearing those thresholds, that could cause a lack of significance. It also is important to note that absorption of micronutrients from diet and supplements are not the same. Again, I think it would be vital to run the same set of analyses with the DII from food intake only to see if your results differ.

[Response]: We appreciate the reviewer’s comments.

As we mentioned earlier, the authors do belive that the nutrients intake from supplements influences the inflammatory potential of the diet, therefore it should be taken into account in the calculation of DII, despite poor bioavailability. The authors suppose that the greatest influence on the lack of statistical significance was due to the high homogeneity of the group and the too small number of the examined women.

I disagree with the authors statement about mentioning biological mechanisms for the findings is not appropriate for this work. One of Sir Austin Bradford Hill's causality criteria which is a staple in epidemiology is biological plausibility. Many DII papers provide plausible explanation for their findings. This helps to drive future research and to open the door for collaborations. You do not need to go into exhaustive detail, because as you say, you didn't investigate that. But, to mention a few mechanisms would be helpful. 

[Response]: We appreciate the reviewer’s comments. The authors presented a few likely biological mechanisms as follows:

„The CRP protein is produced in the liver but can also be synthesized by the placenta and released into the maternal circulation. Perhaps this is one of the mechanisms underlying the increased CRP concentration in pregnant women [29]. Shivappa and colleagues study [30; 31] have shown that  a pro-inflammatory diet is associated with increases in CRP concentrations possibly due to an elevated level of oxygen stress. „ – p. 12, l. 270-274

„IL-6 concentration increases with the duration of pregnancy, with a significantly increased level observed during delivery [12].  Similarly, the expression of IL-10 during pregnancy is variable and depends on the duration of pregnancy. In addition, reduced expression of IL-10 may be an important mechanism associated with the preparation of the body for the onset of labor [8]. Perhaps the pregnant woman body strives to maintain homeostasis in the circulation of the studied cytokines, and excessive stimulation of the inflammatory response through the diet disturbs this homeostasis.” – p.12, l. 283-289

Lastly, the manuscript that was uploaded does not have any indications of where changes were made. The authors noted changes that were made, but unfortunately, it is not possible to easily find such changes. 

[Response]: We appreciate the reviewer’s comments.

The authors realize that finding the introduced corrections was troublesome without indicating their place in the text. As an excuse, the authors would like to explain that, at the editor's request, the revised manuscript was sent as a new manuscript with a new ID. Currently, all corrections are marked red in the text and their place is given in the cover letter.

Reviewer 4 Report

The manuscript has been revised according to the reviewers' comments

Author Response

Reviewer 4

The manuscript has been revised according to the reviewers' comments

The authors are very grateful for the kind review.  All grammatical and stylistic errors were corrected by an American English native speaker.

Round 2

Reviewer 2 Report

The authors answered most of our comments adequately and the manuscript has therefore been improved. However, there are still major issues with the manuscript, namely regarding the use of certain terms relating to causality and in regard to the overall structure of the manuscript. I understand that the authors revised the manuscript, but I still would recommend an additional grammar and syntax editing because many errors remain.

MAJOR COMMENTS

  1. There is an overall lack of structure in the introduction. Please combine the statements that are on the same subject (example: associations between a pro-inflammatory diet and health outcomes) in one paragraph.
  2. Throughout the manuscript (including the abstract), the authors often use the term ‘cause’. I would refrain from using that term, because the analyses performed in the present study can not assess causality. Causality is examined with randomized controlled trials. In the present study, only associations and differences were examined. Rephrase the sections where the word ‘cause’ was used, so that it states what was actually assessed (i.e. associations and differences). The same comment applies for the reference of the study by Sen et al. (line 78), since that study was observational and not an RCT, thus they could not have found any causality.
  3. Similarly, I suggest rephrasing the sentences that states that the ‘use of a pro-inflammatory diet […]’. People don’t ‘use’ diets.
  4. I have a problem with the fact that the authors purposely gave nutritional advice to their participants regarding their diet. It interferes with the observational design of their study and could have impacted their results. This should be mentioned as an important limitation of the study.
  5. The authors discuss the fact that the study of the inflammatory potential of the diet is a promising tool, however their results do not really align with that. It would be important to mention that the negative associations they observed between the DII and IL-6 are not in line with the existing literature and are not 'logical'. Also, since the DII was developed by reviewing studies in non pregnant adults, thus the DII may not apply to pregnant women. This should be mentioned clearly in the discussion.

MINOR COMMENTS

  1. Abstract. Line 28, include the p-value at the end of the sentence regarding DII score across trimesters.
  2. Abstract, results and discussion. The authors mention several times that there were no differences in IL-6, IL-10 and CRP levels « with the exception of CRP level in the second trimester in women with complicated pregnancy (subgroup with DII < median vs subgroup with DII > median). » Could the authors add what subgroup had higher CRP levels, instead of just mentioning how the women were divided in subgroups. Please ensure that that specification is made every time that result is mentioned.
  3. Introduction. Line 51. ‘[…] changes in cytokine concentrations’. Please specify if those changes are increases or decreases.
  4. Introduction. I believe some editing is necessary for the paragraphs, it seems like there are too many short paragraphs in the introduction rather than clear sections.
  5. Introduction. Line 80. Please replace ‘fight’ by ‘counter’ like in the Abstract.
  6. Methods. Lines 101-103. What do the authors mean by ‘food errors’? Please specify and give example of those errors.
  7. Table 1. Please indicate the number of women in each stages of pregnancy in the characteristics table.
  8. Discussion. Line 250. Why use the term ‘The pregnant ones’, when the study only included pregnant women? Please use ‘pregnant women’ instead of ‘the pregnant ones’.
  9. Discussion. Lines 256-257. « High group homogeneity in terms of education, age and the group's residence could contribute to such results. » Please explain this statement... it looks like it comes out of nowhere. Do the authors have literature to support it? If so, it should be included and further explained.
  10. Discussion. Line 294. « The study of the association of the inflammatory potential of the diet using the DII with the concentration […]. »
  11. Conclusions. The conclusion should include implications for clinical practice, what do the results mean for the health and follow-up of pregnant women?

Author Response

Reviewer 2

Comments and Suggestions for Authors

The authors answered most of our comments adequately and the manuscript has therefore been improved. However, there are still major issues with the manuscript, namely regarding the use of certain terms relating to causality and in regard to the overall structure of the manuscript. I understand that the authors revised the manuscript, but I still would recommend an additional grammar and syntax editing because many errors remain.

[Response]: We appreciate the reviewer’s comments. The authors revised the manuscript in accordance with major and minor comments made by the reviewer. All corrections made in the manuscript text are marked in red as well as the place for corrections is indicated. All manuscript text has been checked by an American English native speaker one again.

MAJOR COMMENTS

  1. There is an overall lack of structure in the introduction. Please combine the statements that are on the same subject (example: associations between a pro-inflammatory diet and health outcomes) in one paragraph.

[Response]: We appreciate the reviewer’s comments. The authors edited the Introduction section in accordance with the reviewer's instructions. Short paragraphs covering a related topic have been combined p. 2 l. 58-64; p.2 65-71; p.2 l.72-79 and the obesity paragraph moved to the end of the Introduction section and merged the paragraph on obesity-DII relationship p. 2 l. 80-83

  1. Throughout the manuscript (including the abstract), the authors often use the term ‘cause’. I would refrain from using that term, because the analyses performed in the present study can not assess causality. Causality is examined with randomized controlled trials. In the present study, only associations and differences were examined. Rephrase the sections where the word ‘cause’ was used, so that it states what was actually assessed (i.e. associations and differences). The same comment applies for the reference of the study by Sen et al. (line 78), since that study was observational and not an RCT, thus they could not have found any causality.

[Response]: We appreciate the reviewer’s comments. The authors admit that the use of the word "cause" was a mistake and was replaced with the word "affect" – p. 1 l.29; p.2 l.77; p.11 l.229

  1. Similarly, I suggest rephrasing the sentences that states that the ‘use of a pro-inflammatory diet […]’. People don’t ‘use’ diets.

[Response]: We appreciate the reviewer’s comments. The authors rewrite the sentence as follows:

„There are also reports that suggest an adverse effect of a diet with pro-inflammatory potential in pregnancy on the occurrence of obesity in offspring in early childhood [28].” p.12 l.272-274

  1. I have a problem with the fact that the authors purposely gave nutritional advice to their participants regarding their diet. It interferes with the observational design of their study and could have impacted their results. This should be mentioned as an important limitation of the study.

[Response]: We appreciate the reviewer’s comments. The authors have completed the study limitations as follows:

„Furthermore, the dietary advice given to study participants may have impact on the obtained results.” P.12 l. 320-321

The authors would like to mention, however, that the nutritional advice given to pregnant women was based on a nutritional interview collected during the visit and concerned the next trimester of pregnancy. Therefore, nutritional education had no effect on the diet in the first trimester in any of the studied women, it could affect the change of eating habits / choices only in 21 women in the second trimester and 25 in the third trimester.

  1. The authors discuss the fact that the study of the inflammatory potential of the diet is a promising tool, however their results do not really align with that. It would be important to mention that the negative associations they observed between the DII and IL-6 are not in line with the existing literature and are not 'logical'. Also, since the DII was developed by reviewing studies in non pregnant adults, thus the DII may not apply to pregnant women. This should be mentioned clearly in the discussion.

[Response]: We appreciate the reviewer’s comments. The authors have completed the Discussion section as follows:

„The demonstrated correlation for IL-6 is surprising and different from the reports of other authors.” P.12 l.292-294

„The results of the studies conducted so far are not unequivocal. On the one hand, modulating the inflammatory process through a properly balanced diet seems to be a promising tool in the fight against excessive inflammatory response during pregnancy on the other hand, some studies have shown no association between a pregnant woman's diet and her body's immune response. It should also be remembered that the DII has been validated on the non pregnant adult population and may not apply to pregnant women.” P.12 l.308-312

MINOR COMMENTS

  1. Abstract. Line 28, include the p-value at the end of the sentence regarding DII score across trimesters.

[Response]: We appreciate the reviewer’s comments. The authors have completed the Results section in Abstract as follows:

„With the development of pregnancy, the DII score slightly decreased in subsequent trimesters: -1.78 in the first trimester, -2.43 in the second trimester and -2.71 in the third trimester (p=0.092).”´p.1 l. 26-28

  1. Abstract, results and discussion. The authors mention several times that there were no differences in IL-6, IL-10 and CRP levels « with the exception of CRP level in the second trimester in women with complicated pregnancy (subgroup with DII < median vs subgroup with DII > median). » Could the authors add what subgroup had higher CRP levels, instead of just mentioning how the women were divided in subgroups. Please ensure that that specification is made every time that result is mentioned.

[Response]: We appreciate the reviewer’s comments. The authors changed the description of the results

„Independent of the trimester of pregnancy and the occurrence of pregnancy complications, the DII score didn’t have affect on differences in the serum concentrations of IL-6, IL-10 and CRP with the exception of CRP level in the second trimester in women with complicated pregnancy (subgroup with DII < median had a lower CRP level than subgroup with DII > median).” P.1 l. 28-32; p.7 l. 189-193; p.11 l. 228-232

  1. Introduction. Line 51. ‘[…] changes in cytokine concentrations’. Please specify if those changes are increases or decreases.

[Response]: We appreciate the reviewer’s comments. The authors have improved this part Introduction section as follows:

„The first stage of pregnancy is associated with inflammation, which is required for blastocyst implantation, than pro-inflammatory cytokine level increase. The second stage is characterized by an elevated anti-inflammatory cytokine level that is necessary for fetal growth. In the third stage, there is a change to an inflammatory immune state with high pro-inflammatory cytokine level, which is necessary for labour and delivery [6-9].” p.2 l.53-57

  1. Introduction. I believe some editing is necessary for the paragraphs, it seems like there are too many short paragraphs in the introduction rather than clear sections.

[Response]: We appreciate the reviewer’s comments. The authors edited the Introduction section in accordance with the reviewer's instructions. Short paragraphs covering a related topic have been combined p. 2 l. 58-64; p.2 65-71; p.2 l.72-79 and the obesity paragraph moved to the end of the Introduction section and merged the paragraph on obesity-DII relationship p. 2 l. 80-83

  1. Introduction. Line 80. Please replace ‘fight’ by ‘counter’ like in the Abstract.

[Response]: We appreciate the reviewer’s comments. The authors have improved this sentence as follows:

„An anti-inflammatory diet can be a promising way to counter an excessive inflammatory response in pregnancy, especially in obese pregnant women.” p.2 l.78-80

  1. Methods. Lines 101-103. What do the authors mean by ‘food errors’? Please specify and give example of those errors.

[Response]: We appreciate the reviewer’s comments. The authors presented examples of nutritional errors in the Discussion section.

„Irregular meals were observed among the most common food errors in the studied pregnant women as well as too low consumption of vegetables and fruit, fish and milk and too high consumption of highly processed products (data not shown).” p.11 l. 255-258

  1. Table 1. Please indicate the number of women in each stages of pregnancy in the characteristics table.

[Response]: We appreciate the reviewer’s comments. The authors indicated the number of women in each stages of pregnancy in the characteristics table – Table 1 p. 2 l.107

  1. Discussion. Line 250. Why use the term ‘The pregnant ones’, when the study only included pregnant women? Please use ‘pregnant women’ instead of ‘the pregnant ones’.
  1. [Response]: We appreciate the reviewer’s comments. The authors rewrite the sentence as follows:

„The pregnant women in our study were informed on each visit about the food errors that they made and obtained nutritional advice appropriate to the pregnancy stage.” p. 11 l.254-255

  1. Discussion. Lines 256-257. « High group homogeneity in terms of education, age and the group's residence could contribute to such results. » Please explain this statement... it looks like it comes out of nowhere. Do the authors have literature to support it? If so, it should be included and further explained.

[Response]: We appreciate the reviewer’s comments. The authors have improved this part Discussion section as follows:

”Research suggest that older age, higher education level and socioeconomic status are associated with healthier dietary patterns among pregnant women [29,30,31]. This information indicates that within a homogeneous group, the food choices will be very similar, which may be reflected in the DII score.” p. 11 l.264-267

Supporting literature:

Stråvik, M., Jonsson, K., Hartvigsson, O., Sandin, A., Wold, A. E., Sandberg, A. S., & Barman, M. (2019). Food and nutrient intake during pregnancy in relation to maternal characteristics: Results from the NICE Birth Cohort in Northern Sweden. Nutrients, 11(7), 1680.

Hillier, S. E., & Olander, E. K. (2017). Women's dietary changes before and during pregnancy: A systematic review. Midwifery, 49, 19-31.

Wesołowska, E.; Jankowska, A.; Trafalska, E.; Kałużny, P.; Grzesiak, M.; Dominowska, J.; Hanke, W.; Calamandrei, G.; Polańska, K. Sociodemographic, Lifestyle, Environmental and Pregnancy-Related Determinants of Dietary Patterns during Pregnancy. Int. J. Environ. Res. Public Health 2019, 16, 754.

  1. Discussion. Line 294. « The study of the association of the inflammatory potential of the diet using the DII with the concentration […]. »

[Response]: We appreciate the reviewer’s comments. The authors rewrite the sentence as follows:

„The study of the association of the inflammatory potential of the diet using the DII with the concentration of inflammatory mediators as well as on the course of pregnancy among women expecting a baby is an issue that the scientific community has only been dealing with for the last several years.” p. 12 l. 305-307

  1. Conclusions. The conclusion should include implications for clinical practice, what do the results mean for the health and follow-up of pregnant women?

[Response]: We appreciate the reviewer’s comments. The authors have improved this part of Conclusions section as follows:

„Future research is warranted to explore whether diet is a significant contributor to maternal overall inflammation and to demonstrate the potential impact of dietary interventions during pregnancy on maternal and offspring health.” p. 13 l. 331-333

Reviewer 3 Report

I think the reviewers for providing feedback. I was unaware of the fact that this was submitted as a new manuscript. I can understand much better the rationale for changes and can see the changes now. 

Although, I would have conducted the supplement with and without analysis in this paper, I can see the argument the authors made and am satisfied with their response. 

Minor comments:

I believe the word rapture should be rupture on line 62.

On line 199, this doesn't quite make sense to me: "maternal circulating two cytokines and CRP"

On line 228, "waek" should be "weak".

Author Response

Reviewer 3

Comments and Suggestions for Authors

I think the reviewers for providing feedback. I was unaware of the fact that this was submitted as a new manuscript. I can understand much better the rationale for changes and can see the changes now. 

Although, I would have conducted the supplement with and without analysis in this paper, I can see the argument the authors made and am satisfied with their response. 

The authors are very grateful for the kind review. 

Minor comments:

I believe the word rapture should be rupture on line 62.

[Response]: We appreciate the reviewer’s comments. The authors rewrite the sentence as follows:

”In some cases, higher concentrations of some inflammatory markers are associated with pregnancy complications  such as: premature fetal membrane rupture, premature labor, hypertension and preeclampsia, considered to be one of the most common causes of prematurity [10-13].” p.2 l.61-64

On line 199, this doesn't quite make sense to me: "maternal circulating two cytokines and CRP"

[Response]: We appreciate the reviewer’s comments. The authors rewrite the sentence as follows:

„We investigated the correlation between maternal serum IL-6, IL-10 and CRP concentration and the DII score (in subgroups: above and below the median DII score) in particular trimesters among women with normal pregnancy.” p. 7 l.202-204

On line 228, "waek" should be "weak".

[Response]: We appreciate the reviewer’s comments. The authors rewrite the sentence as follows:

„In the first and third trimesters, there was a weak but significant positive correlation between the DII score and CRP concentration.” p.11 l.231-232
